# ROYAL SOCIETY
# OPEN SCIENCE

## Comment

**Author for correspondence:**
Gregory Paul
e-mail: GSP1954@aol.com

# Comment on Brocklehurst *et al.*

## Gregory Paul

3100 St Paul St 604, Baltimore, MD 21218, USA

GP, 0000-0002-4493-6092

Brocklehurst *et al.* [1] present additional evidence that the osteological features of non-avian theropod dinosaurs are incompatible with the presence of the crocodilian-like liver pump respiratory complex that had been proposed by some palaeobiologists [2,3]. The latter hypothesis is in opposition to the wider consensus that the avian air-sac system first appeared in non-avian dinosaurs. In particular, Brocklehurst *et al.* detail that the strongly corrugated ceiling of the theropod ribcage should have locked their rigid lungs in place, as in birds, while the much smoother ribcage ceiling of crocodilians facilitates the fore-and-aft expansion and contraction of their compliant lungs through corresponding movement of their large livers. In doing so, Brocklehurst *et al.* cite earlier studies to the same effect dating back to 2009 by Schachner *et al.* [4,5].

My concern is that Brocklehurst *et al.* neglected to cite my even earlier work, starting in 2001 [6–8], which as far as I am aware first explicitly and extensively tested the configuration of theropod respiratory apparati by comparing the contrasting topography of dinosaur versus crocodilian ribcage ceilings in the technical literature. The first paper, the subsequent academic book appendix and the later book chapter included comprehensive text backed by the first detailed comparative illustrations of the pertinent vertebra-rib head articulations. These studies were conducted with the direct intent of addressing the problematic Ruben *et al.* hypothesis [2,3], first published in 1997, that some dinosaurs had crocodilian-like rather than avian-like breathing complexes. Those researchers have since failed to plausibly refute my arguments, a task made increasingly difficult by subsequent and increasingly sophisticated research on the subject. Yet this precedent is not noted by Brocklehurst *et al.*

They are not the first to fail to cite some of my innovative arguments on the subject of archosaur respiration. It is especially pertinent to this discussion that while the two Schachner *et al.* analyses cite my studies, they too did not explicitly relate that I had previously examined the contrasts between bird and crocodilian ribcage morphologies and came to similar conclusions. Perhaps that failing to cite the precedent literature contributed to Brocklehurst *et al.*'s ignorance of who first considered and refuted the Ruben *et al.* hypothesis.

I regret that I did not cite the Schachner *et al.* 2009 and 2011 papers in my 2012 discussion of the subject because I was not aware of them. Had I known of the studies I would have included them, the more to support the anatomical comparative hypothesis I originated.

Hopefully, this note will cause those in the future working on this aspect of archosaur evolution to cite the precedent literature properly, whether or not they agree with it.

Data accessibility. There is no data specific to this commentary.
Competing interests. I declare no competing interests.
Funding. There was no financial support for this commentary.

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
