## [Reviewer comments · Royal Society Open Science]

Review History

Decision letter (RSOS-181872)

23-Jan-2019

Dear Dr Paul:

On behalf of the Editors, I am pleased to inform you that your Manuscript RSOS-181872 entitled "Comment on Brocklehurst *et al.*" has been accepted for publication in Royal Society Open Science subject to minor revision in accordance with the referee suggestions. Please find the referees' comments at the end of this email.

The Subject Editor have recommended publication, but also suggest some minor revisions to your manuscript. Therefore, I invite you to respond to the comments and revise your manuscript.

- Ethics statement

- Data accessibility

If you wish to submit your supporting data or code to Dryad (<http://datadryad.org/>), or modify your current submission to dryad, please use the following link:
<http://datadryad.org/submit?journalID=RSOS&manu=RSOS-181872>

- Competing interests

- Authors' contributions

- Acknowledgements

- Funding statement

Because the schedule for publication is very tight, it is a condition of publication that you submit the revised version of your manuscript before 01-Feb-2019. Please note that the revision deadline will expire at 00.00am on this date. If you do not think you will be able to meet this date please let me know immediately.

on behalf of Professor Kevin Padian (Subject Editor)
openscience@royalsociety.org

Editor's Comments to Author ():

The Subject Editors has recommended some changes to the Comment article. Please see the attached word document for details. If you have any objections to the proposed changes, please outline them in your 'response to referees' document.

Author's Response to Decision Letter for (RSOS-181872)

See Appendix A.

Decision letter (RSOS-181872.R1)

07-Feb-2019

Dear Dr Paul,

I am pleased to inform you that your manuscript entitled "Comment on Brocklehurst et al." is now accepted for publication in Royal Society Open Science.

on behalf of Professor Kevin Padian (Subject Editor)
openscience@royalsociety.org

Appendix A

All reviewer's corrections are accepted, and the reference correction has been made. Editors can decide if items concerning animal ethics and fieldwork should be deleted.